



**Continental anthropogenic primary particle number**
**emissions**
**P. Paasonen[1,2,3], K. Kupiainen[2,3], Z. Klimont[2], A. Visschedijk[4], H. A. C. Denier**
**van der Gon[4], M. Amann[2]**
[1]{University of Helsinki, Helsinki, Finland}
[2]{International Institute for Applied Systems Analysis, Laxenburg, Austria}
[3]{Finnish Environment Institute, Helsinki, Finland}
[4]{TNO, the Netherlands}
Correspondence to: P. Paasonen (pauli.paasonen@helsinki.fi)
**Abstract**
Atmospheric aerosol particle number concentrations impact our climate and health in ways
different from those of aerosol mass concentrations. However, the global, current and future,
anthropogenic particle number emissions and their size distributions are so far poorly known.
In this article, we present the implementation of particle number emission factors and the
related size distributions in the GAINS model. This implementation allows for global
estimates of particle number emissions under different future scenarios, consistent with
emissions of other pollutants and greenhouse gases. In addition to determining the general
particulate number emissions, we also describe a method to estimate the number size
distributions of the emitted black carbon. The first results show that the sources dominating
the particle number emissions are different to those dominating the mass emissions. The
major global number source is road traffic, followed by residential combustion of biofuels
and coal (especially in China, India and Africa), coke production (Russia and China), and
industrial combustion and processes. The size distributions of emitted particles differ across
the world, depending on the main sources: in regions dominated by traffic and industry, the
number size distribution of emissions peaks in diameters range from 20 to 50 nm, whereas in
regions with intensive biofuel combustion and/or agricultural waste burning, the emissions of
particles with diameters around 100 nm are dominant. In the baseline (current legislation)
scenario, the particle number emissions in Europe, Northern and Southern Americas,
Australia, and China decrease until 2030, whereas especially for India, a strong increase is





estimated. The results of this study provide input for modelling of the future changes in
aerosol-cloud interactions as well as particle number related adverse health effects, e.g., in
response to tightening emission regulations. However, there are significant uncertainties in
these current emission estimates and the key actions for decreasing the uncertainties are
pointed out.
**1. Introduction**
Aerosol particles affect both our health and the climate in many ways. These effects depend
partly on the composition of the particles and partly on their sizes and concentrations (WHO,
2013; Stocker et al., 2013). Furthermore, different effects are linked to different metrics of
concentration –mass and number. Because of the cubic relation between particle mass and
diameter, $d_p$, it is common that these metrics of concentration are dominated by particles with
very different sizes. Clearly the largest numbers of particles are typically observed in the size
range of ultrafine particles (UFP) with $d_p < 0.1$ µm or the smaller end, roughly $< 0.3$ µm, of
fine particles (FP, here $0.1 – 2.5$ µm), whereas the mass concentration depends mostly on the
larger and heavier, but typically fewer FP, with $d_p > 0.3$ µm (see Fig. 1 for schematic
representation). Because the particles in different size ranges originate from different sources
and atmospheric processes impact them differently, the particle number (PN) concentrations
are often not well correlated with particle mass concentrations (PM, e.g. $PM_{2.5}$ describing
mass concentration of particles with $d_p < 2.5$ µm).
According to WHO (2013), there is increasing epidemiological evidence on the association
between short-term exposures to ultrafine particles and cardiorespiratory health, as well as the
health of the central nervous system. Clinical and toxicological studies indicate that the
mechanisms that cause the health effects of ultrafine particles are (in part) not the same as due
to larger particles, such as $PM_{2.5}$ or $PM_{10}$ (WHO, 2013). Also the climate effects of aerosol
particles depend on their size (Stocker et al., 2013). All particles can, depending on their
chemical composition, either absorb solar radiation (mainly black carbon aerosol) or scatter it
partly back to space. In addition to these so called aerosol-radiation interactions, the particles
with diameter close to or over 0.1 µm can act as cloud condensation nuclei (CCN), i.e. they
can form cloud droplets when the air mass moves upwards and cools down. Since the clouds
efficiently reflect solar radiation back to space, these aerosol-cloud interactions have a
significant cooling effect on our climate. One of the problems in assessing the total radiative
forcing of aerosols is the non-linear relationships of these different interactions, e.g.



depending on the initial sizes and atmospheric growth of black carbon particles, their
warming effect due to light-absorption can be neglected, either partly or entirely, by their
ability to form cloud droplets. The future reductions in anthropogenic emissions of aerosol
and their precursors have been estimated to accelerate global warming (Arneth et al., 2009;
Makkonen et al., 2012; Westervelt et al., 2015). However, the changes in aerosol-cloud
interactions have been so far either ignored or assessed by converting the mass emissions to
number emissions, which leads to incorrect assumptions in case the size distributions of
emitted particles change.
The ultrafine and fine particles originate from a number of sources and atmospheric
processes. New particle formation (i.e. nucleation) both in the atmosphere and in the
combustion plumes produces particles with diameters below 2 nm (0.002 μm) from vapours
such as sulphuric acid, organic vapours and nitrogen containing bases. Somewhat larger UFP
particles in nucleation mode are formed, e.g., in nucleation processes occurring already before
the trace gases are emitted to the atmosphere and thus producing cores for cooling vapours to
condense on (e.g. Rönkkö et al., 2007; Lähde et al., 2010). Black carbon, i.e. soot particles,
formed in flames by agglomeration of cyclic carbon molecules and emitted often with a
coating of condensed organic or inorganic vapours, are also partly in UFP size range (< 0.1
μm), but their size distribution extends to FP size range. FP are emitted also from other
thermal sources, as well as from mechanical sources like dust resuspension, wear,
fragmentation and suspension of biological matter. Fine particles are also formed from
ultrafine particles by growth via atmospheric condensation of anthropogenic and biogenic
organic compounds, sulphuric acid and nitrates on the particle's surface. Biogenic
condensation growth of UFP is a significant contributor to fine particle number
concentrations. It has been estimated that out of the total number of fine particles over the
European continent, roughly 50 % have been formed through growth of UFP by condensed
biogenic organic vapours (Paasonen et al., 2013a).
The legislation on aerosol emissions and concentrations is based on particle mass, mainly due
to the well-established knowledge on the correlation of $PM_{2.5}$ and adverse health effects (Pope
et al., 2002, 2009). However, the increasing evidence of the adverse health impacts of UFP as
well as the unresolved significant uncertainties on the aerosol-climate effects due to aerosol-
cloud interactions, require more attention to the anthropogenic particle number emissions.
The mass emissions cannot be directly converted to number emissions, because the ratio of





mass and number emissions depends greatly on the size distribution of emitted particles.
Additionally, because the main removal mechanism of ultrafine aerosol particles in the
atmosphere is their coagulation to larger particles (e.g. Kerminen et al., 2001), a decrease in
e.g. $PM_{2.5}$ emissions might even increase PN concentrations (Pirjola et al., 2015).
In global climate modelling work, the number emissions are typically extracted from mass
emissions applying constant factors and size distributions for different highly aggregated
source sectors (e.g., traffic, biomass burning, power generation, etc.). This approach can be
used to produce future scenarios also for number emissions and their size distributions. An
example of such an approach is the widely applied emission database, the AeroCom project
(Dentener et al., 2006), in which the size distributions are fixed and averaged over wide
variety of different sources under the main sectors. Thus, the changes in technology and fuels
are reflected in number emissions through a linear dependence between mass and number
emissions, since the size distribution is assumed to remain the same. On the other hand, the
aerosol number emissions and their size distributions with information on different emission
abatement techniques have been studied lately resulting in a size-resolved European particle
number emission inventory (Denier van der Gon et al., 2010; 2013; 2014; Kulmala et al.,
2011) which has been tested in several UFP modelling exercises (e.g. Fountoukis et al., 2012;
Kukkonen et al., 2015). Emission inventories are not directly applicable for estimating the
future trends in emissions as they are based on available statistics, which generally lag several
years behind present day. However, in combination with projections of activity data and
assumptions about penetration of control technologies a present day inventory can form a
starting point for projected future emissions.
Here we describe and present the first results of the implementation of aerosol number
emission factors and their size distribution to the global emission scenario model GAINS
(Greenhouse gas – Air pollution Interactions and Synergies; Cofala et al, 2007; Amann et al,
2011) developed at IIASA (International Institute for Applied Systems Analysis, Austria).
The implementation of these factors in the GAINS-Europe model, describing only European
emissions, was published in a IIASA report (Paasonen et al., 2013b).
We also estimate emissions and size distribution of the black carbon containing particles and
the black carbon cores in them. The GAINS model has a more detailed technological structure
than many available inventories and thus we are able to estimate the implications of future
abatement technology changes on number emissions and size distributions. GAINS has been



previously applied to analyse the effect of emission abatement policies and other factors
affecting the emissions in terms of traditional air pollutants, including particle mass, and
greenhouse gases. The results from GAINS model are widely applied for air quality
legislation especially in the European Union (Amann et al., 2013). With the implementation
of aerosol number emission factors to GAINS, the future particle number emissions can be
estimated in a consistent manner with other air pollutants and greenhouse gases. This
information can be used for estimating the effects of emission regulations and technological
improvements on the health effects of ultrafine particles and on aerosol-climate effects in
future decades, as well as for planning particle number emission measurements for the
sources that are so far not well enough reported.

## 12  2.  Methods

### 13  2.1.  The GAINS model

The GAINS (Greenhouse gas – Air pollutant Interactions and Synergies) model (Amann et
al., 2011) is an integrated assessment model that brings together information on the sources
and impacts of air pollutant and greenhouse gas emissions and their interactions. GAINS
combines data on economic development, the structure, control potential and costs of
emission sources, the formation and dispersion of pollutants in the atmosphere and an
assessment of environmental impacts of pollution.
GAINS assesses all the main air pollutants and greenhouse gases ($SO_2$, $NO_x$, PM, NMVOC,
$NH_3$, $CO_2$, $CH_4$, $N_2O$, F-gases) with more than 1000 measures to control the emissions to the
atmosphere for each of its nearly 170 regions. Applying built in source-receptor relationships
(developed in collaboration with atmospheric groups running chemical transport models for a
given domain), GAINS identifies the least-cost balance of emission control measures across
pollutants, economic sectors and countries that meet user-specified air quality and climate
targets.
In GAINS, emissions from different sources are calculated with three basic input parameters
(Klimont et al., 2002):
-  Annual activity levels ($A$) in a given sector, corresponding to certain fuels (e.g., fuel
wood used (burned) per year in domestic single house boilers),





-    Shares ($X$) of abatement technologies applied to fuel consumption of the activity (e.g.,

2         improved boilers with accumulation tank, pellet boilers, boilers with electrostatic

3         precipitator, etc.) such that $\sum X = 1$,

-    Emission factors (EF) for each sector-fuel-technology –combination (emissions per unit

5         of activity).

Activity levels $A$ in GAINS are based on the information from international statistics
available from International Energy Agency (IEA), Organisation for Economic Co-operation
and Development (OECD), United Nations (UN) and Food and Agriculture Organization of
the United Nations (FAO), Eurostat, and national statistics. The shares of control technologies
X are derived from published information on national and international legislation, for
example for transport sector from diesel.net, discussions with the national experts, and
scientific publications where similar assessment has been performed. The emission factors EF
are determined from the scientific publications and measurement databases.
The yearly emissions $E$ in region $i$ are calculated as
$E_i = \sum_{ijkm} E_{ijkm} = \sum_{ijkm} A_{ijkm} X_{ijkm} \mathrm{EF}_{ijkm}$ ,                    (1)
where the indices $j$ refer to source sector, $k$ to fuel and $m$ to abatement technology.
Within GAINS, future emissions are estimated for different scenarios of anthropogenic
activities (e.g., energy use), for which shares $X$ of different technology levels for all emission
sources are assumed. Here we present results based on the Current Legislation (CLE) baseline
scenario created in the ECLIPSE project, specifically version 5 of this scenario
(ETP_CLE_v5, Klimont et al., 2016a, 2016b; Stohl et al., 2015).
**2.2.    Particle number emission factors and size distributions**
The determination of emission factors (EF$_{PN}$) for particle number (PN) emissions and particle
size distributions (PSD) is based on the European particle number emission inventory
developed by TNO (Denier van der Gon et al., 2009, Denier van der Gon et al., 2010) during
the EUCAARI project (Kulmala et al., 2011). In that work, as well as here, the PSDs present
the size segregation of the number emissions into size classes, i.e., the proportions $P_i$ of the
total number of emitted particles in each size sector $i$. Thus, the emission factor for a single
size class $i$ is written as



$EF_{PN,i} = P_i EF_{PN},$                                                      (2)
and the $\Sigma P_i = 1$. Values for the proportions $P_i$ are calculated from modal presentations of
PSDs, consisting of one to three lognormal modes.
$EF_{PN}$:s were determined through two alternative ways. For some source sectors, including
traffic and domestic combustion, both $EF_N$:s and PSDs were determined from the literature
directly (these are called hereafter as direct emission factors). For other source sectors, $EF_{PN}$:s
were determined based on $PM_1$ mass emission factors ($EF_{PM1}$) from an earlier version of the
GAINS model (Kupiainen and Klimont, 2004). However, particle number size distributions
are very uncertain in size ranges close to 1 µm, where the number is very small compared to
that of smaller particles. Because deriving an $EF_{PN}$ directly from the $EF_{PM1}$ would make the
$EF_{PN}$ very sensitive to the estimated number of those close to 1 µm particles, emission factors
for PM in the size range 10-300nm ($EF_{PM0.3}$) were first derived from $EF_{PM1}$ based on literature
on emission mass size distributions and particle densities (M. Kulmala et al., 2011, H. Denier
van der Gon et al., 2010). Then, by applying the particle number size distributions from the
literature, the $EF_{PN}$:s consistent with $EF_{PM0.3}$ were resolved. The latter type of emission
factors is called PM-based emission factors, hereafter.
In our analysis, we employ for many source sectors the emission factors and size distributions
provided in the TNO study.  However, for sources that are most important for particle
numbers, such as road transport and wood combustion in the domestic sector, we developed
new emission factors and size distributions in order to better fit in the GAINS model,
especially in terms of the emission abatement technologies within it. The modifications to the
TNO study are described below.
We extended the PSDs in GAINS to cover sizes from electrical mobility diameter ($d_M$) of 3
nm up to aerodynamic diameter ($d_A$) of 1 µm, whereas the particle size range in the TNO
study was from $d_M$ =10 nm to $d_A$ =300 nm. The size range was extended to larger sizes in
order to allow for comparison between the emission factors for particle number and PM1
mass, the latter being determined as the total mass of particles with $d_A \leq 1$ µm. Additionally,
even though the number share of particles larger than 300 nm in all emitted particles is
negligible, large particles are important in some source sectors. The extension towards smaller
diameters was made to provide the whole particle size range for climate model calculations,
but it should be noted that no modes with diameters below 10 nm were introduced. These





extensions of the particle size ranges required recalculation of the $EF_{PN}$:s for source sectors
that were originally based on PM0.3 emission factors, with the formula
$$EF_N = \frac{1}{\rho \sum_i P_i \frac{\pi}{6} d_i^3} R(PM_{0.3}/PM_1) EF_{PM1}, \qquad (3)$$
where $\rho$ is the estimated density of the emitted particles, $P_i$ is the proportion of particles in
size class $i$ out of the total number of emitted particles, $d_i$ is the geometric mean diameter of
the particles in size class $i$, and $R(PM_{0.3}/PM_1)$ describes the ratio of $PM_{0.3}$ and $PM_1$ –masses.
The values for $\rho$, $R(PM_{0.3}/PM_1)$ and PSDs were taken from the TNO analysis, with the
exception of the PSDs mentioned below.
New PSDs were introduced for road transport sources with the highest activities (diesel heavy
duty trucks and busses, both diesel and gasoline light duty trucks and passenger cars), based
on the EU FP7 project TRANSPHORM database (Vouitsis et al., 2013). Additionally, the
emission factors for diesel fuelled road transport were made dependent on the fuel sulphur
content (FSC), based on vehicle-specific FSC dependent emission factors provided by the
Laboratory of Applied Thermodynamics at the Aristotle University of Thessaloniki, which is
responsible also for the TRANSPHORM database. Also $EF_N$:s and PSDs for domestic wood
combustion (including pellet burning and medium size district heating boilers) and for
shipping emissions (fuel sulphur content –dependent EFs and PSDs) were updated (domestic
sector: Gaegauf et al., 2001, Emma Hedberg et al., 2002, L. S. Johansson et al., 2004, C.
Johansson et al., 2008, Kinsey et al., 2009, Lamberg et al., 2011, Bäfver et al., 2011, C.
Boman et al., 2011, Pettersson et al., 2011, Chandrasekaran et al., 2011; shipping: Hobbs et
al., 2000, Sinha et al., 2003, Petzold et al., 2008, Murphy et al., 2009, Moldanova et al., 2011,
Diesch et al., 2013), as well as for two stroke vehicles in road transport (Ntziachristos et al.,
2005, Etissa et al., 2008). New PSD was introduced also for flaring in gas and oil industry
(Canteenwalla et al., 2006). The $EF_{PN}$ for tire wear, previously based on $EF_{PM0.3}$, was replaced
with a direct PN emission factor (Dahl et al., 2006).
We note that many of the measured $EF_{PN}$:s and PSDs are not representing the particles which
either have diameters below 10 nm or are volatile in temperatures above typical atmospheric
temperatures. Thus, it is likely that in the current set of emission factors the nucleation mode
particles ($d_p < 20$ nm), which are formed from vapour molecules during their initial cooling
when introduced to the atmosphere, is largely overlooked.





## 2.3. Black carbon size distribution estimates
In addition to determining the emission factors and size distributions for total particle number
emissions, we also made estimates for black carbon emission size distributions. Two different
size distributions were determined, one for the whole particles in black carbon mode ($BC_{mode}$)
and one for the black carbon cores of these particles ($BC_{core}$).
The division of emitted particles to black carbon containing particles and other particles was
made depending on the source of particles and the geometric mean diameters of the number
size modes of the emitted particles. Naturally, only the combustion related sources were
considered to produce black carbon. Of the combustion sources, only the modes with
geometric mean diameters (GMD) equal to or above 50 nm were assumed to be black carbon
modes, because the agglomeration in BC formation produces a roughly lognormal size
distribution and would not form particles with diameters especially in the lower end of the
size ranges of the modes with GMD below 50 nm (Sorensen et al., 1996; Kholghy et al.,

14 2013).

The size distribution of the black carbon cores in the black carbon containing particles was
calculated with two combinations of assumptions. In both it was assumed that all the BC
mode particles (defined as above) have a black carbon core and that both the core and the
particle are spherical. The difference was that in one calculation we assumed that there is only
organic carbon (OC) condensed on the BC core, and in the other calculation that all $PM_1$
additional to BC is condensed onto this core. The shares of BC, OC and other $PM_1$ were
defined with mass emission factors for BC, OC and $PM_1$ in GAINS. A further, simplified
assumption was made that the shares of BC and OC (or BC and other $PM_1$, when considered
as an additional condensed matter) were the same in all BC containing particles regardless
their size. This might slightly overestimate the share of condensed matter in BC mode for the
sources in which there is significant non-BC mode (with GMD<50 nm). The geometric mean
diameters of the BC-cores were derived simply from these assumptions based on the mass
emission factors and BC-mode geometric mean diameter $GMD_{BCmode}$. For the case of only
OC condensing on the particles the geometric mean diameter of the core was
$$GMD_{BCcore1}=GMD_{BCmode}\times\left(\frac{EF_{BC}}{EF_{BC}+EF_{OC}}\right)^{1/3} \qquad (4)$$
and, for the case of all $PM_1$ except BC being formed through condensation





$GMD_{BCcore2} = GMD_{BCmode} \times \left( \frac{EF_{BC}}{EF_{PM1}} \right)^{1/3}.$        (5)
**2.4.**    **Uncertainties**
In the results presented in Section 3 we have not depicted error bars or shown other
illustration of uncertainties. The major sources of uncertainties are mentioned in text within
Sect. 3, and discussed in more detail in Sect. 4.
**3.**      **Results**
The calculated aerosol number emissions in 2010 were dominated by ultrafine particles,
which contributed to total PN emissions by about 80 %. However, emissions from different
sources varied in terms of particle size, which is presented in the lower panel of Fig. 2 as the
division of number emissions to UFP and FP size ranges in each source sector. The upper
panel of Fig. 2 shows the shares of different sources in the global anthropogenic continental
total particle number emissions, number emissions of ultrafine particles (UFP, $d_p < 0.1\,\mu m$)
and FP ($d_p > 0.1\,\mu m$), as well as mass emissions of particles with $d_p < 1\,\mu m$ ($PM_1$), all for year
2010. The main source of UFP was road transport, corresponding to 40 % of UFP emissions
and thus being the largest contributor to total aerosol particle number emissions. Also power
production contributed to the UFP emissions with 20 % share, mainly due to emissions from
coke production, and residential combustion with 17 % share. In FP size range, the shares of
residential combustion and road transport were quite similar, roughly 30 % each, whereas the
mass emissions of particles with diameters below 1 μm ($PM_1$) were clearly dominated by
residential combustion (> 50 %). These differences indicate the need for assessing the size
segregated number emissions of aerosols in addition to mass emissions, in order to better
understand their role in atmospheric processes as well as their climate and health effects. It is
also important to notice that there is most probably more difference between number
emissions and $PM_{2.5}$ mass emissions (which is often the regulated and monitored quantity)
than between number emissions and $PM_1$ emissions.
**3.1.**     **Overall emissions in different parts of the world**





Total annual aerosol number emissions and their current trend for different continents, with
Eurasia divided to major countries and the rest of Europe and Asia, are depicted in Figure 3.
The future trend is based on the current legislation baseline scenario (ETP_CLE_v5, Stohl et
al., 2015). In 2010, China emitted clearly the most aerosol particles, followed by Asia (excl.
China, India and Russia) and Europe (excl. Russia). However, the actions determined in the
current legislation scenario resulted in decrease of emissions in China, as well as in Europe,
North- and South-America. On the contrary, especially in India, but also in Russia, Asia and
Africa, the increase in activities seem to offset the benefits of more stringent legislation. The
global sum of continental anthropogenic emissions is expected to decrease from 2010 to 2020
by roughly 15 % (from $1.5 \times 10^{28}$ to $1.3 \times 10^{28}$ particles/year), but remains quite constant from
2020 to 2030.
**3.2.    Main aerosol number sources in 2010 and expected changes until 2030**
The aerosol number emissions were dominated by road transport in Europe, Northern and
Southern Americas, Asia and Australia in 2010 (blue bars in Figure 4). In Africa and India the
emissions from residential combustion were the main sources together with road transport,
whereas in Russia, the emissions from industrial processes, road transport and non-road
transport were on a similar level. In China, the major source sector for particle number
emissions was power production, followed by residential and industrial combustion
emissions. In general, it should be noted that with the current set of emission factors the
uncertainties are lesser in Northern America and Europe, where most of the applied emission
factor measurements are made (more in Section 4).
In the following subsections (3.2.1.-3.2.5.), we discuss separately the major sources of aerosol
number emissions and their predicted changes from 2010 to 2030. In these subsections, the
percentages given for the shares of different sources refer to emissions in 2010, if not stated
otherwise.
**3.2.1. Power production emissions**
The dominance of the power production emissions in China was caused by the emissions
from coke production, which accounted for 95 % of Chinese power production emissions in
2010. Also the significant contributions of power production to emissions in Russia and India
were caused by coke production (88 % and 79 %, respectively).





The coke production emissions in China were estimated to decrease over 50 % from 2010 to
2020, whereas in India and Russia coke production emissions were predicted to increase by
200 % and 70 %, respectively. The decrease in Chinese emissions resulted mainly from large
scale replacement and closure of small inefficient coke ovens with modern installations, often
equipped also with measures to capture and remove dust emissions, which offsets the 20 %
increase in activity level.  For India and Russia, changes in abatement technology shares did
not take place in the applied CLE-scenario, and thus the changes were due to increased
activity levels only.
However, the coke production emissions are subject to significant uncertainties. Additionally,
the power production emissions from (coal-fired) power plants are not dependent on the
sulphur removal technologies or sulphur contents of the fuels, but only on particle removal
technologies. This may cause underestimation in power plant emission estimates (from other
sources than coke production) in many parts of the world. Thus, the presented results on
power production emissions have to be considered as preliminary estimates. It seems obvious
that coke production causes at least a significant part of the aerosol number emissions in
question, but the future trends especially in China are very uncertain, depending on the rate of
activity level increase and overall emission factor decrease due to improving technology. The
uncertainties are discussed in more detail in Sect. 4.

### 20   3.2.2. Residential combustion

Residential combustion was a significant source of particles, especially in China, India and
Africa. All these emissions originated mainly from cooking stoves, but used fuels varied. In
India, firewood, agricultural residues and coal contributed each by a share of 25 % or more to
the residential combustion emissions, and also dung combustion had a share of over 10 %.  In
China 64 % of the emissions originated from coal combustion, roughly 24 % from
combustion of agricultural residues and only 7 % from firewood combustion, whereas in
Africa 85 % of emissions came from firewood combustion (activity levels for dung
combustion are available only for India). The uncertainties related to residential combustion
emissions are discussed in Section 4.
In India and Africa the residential combustion emissions were expected to increase slightly
due to the increase in the activity levels. On the other hand, the emissions from residential
combustion in cooking stoves in China were estimated to decrease by 25-30 % per decade due





to the reduced coal use in residential sector which results in an overall decrease in residential
combustion emissions in China.

### 3.2.3. Industrial combustion and processes

Industrial combustion was estimated to contribute significantly to the total aerosol number
emissions in China and India, and the emissions from industrial processes were notable in
Russia and India. In China, the industrial combustion emissions were dominated by cast iron
production (75 % of industrial combustion emissions in 2010) and cement production (10 %),
whereas in India the cement production contributed to the industrial combustion emissions by
50 % and cast iron production by less than 10 %. It is notable that in India 20 % of industrial
combustion emissions were related to biomass fuel combustion.
Of industrial processes, the main source of particle number emissions was estimated to be
basic oxygen furnaces, producing over 80 % of Indian and 50 % of Russian emissions. In
Russia the other main sources were primary aluminium production (17 %), open hearth
furnaces (16 %) and electric arc furnaces (13 %), the latter contributing by 13 % also to
Indian industrial processes emissions.
For all industrial emissions, PM-based emission factors were applied. Thus, the differences in
PN emission factors for different emission abatement technologies are not expected to be
fully consistent (see Sect. 4).

### 3.2.4. Traffic emissions

The emissions from traffic were the major source of aerosol particles in most parts of the
world in 2010. This was the case especially in Western countries and Asia excluding China,
India and Russia. Interestingly, even though the total consumption of fuels in road traffic was
highest in Northern America (42 000 PJ/year compared to 31 000 PJ/year in Asia and 27 000
PJ/year in Europe) the calculated emissions were the highest in Asia and the lowest in N-
America. The low emissions in Northern America were due to much smaller percentage of
diesel vehicles than in Europe, whereas the high emissions in Asia were due to *i*) the
significant share of (diesel) fuel having higher sulphur content than in Europe and N-
America, and *ii*) the smaller proportion of vehicles with new emission abatement
technologies.



Based on the measurements collected by Vouitsis et al. (2013), applied for PN emission
factors in the GAINS model, the tightening regulation on particle mass emissions decreased
drastically the number emissions, as well. This lead to a major decrease in European, N-
American and Australian emissions from 2010 to 2030, as can be seen in Figs. 3 and 4.
Additionally, traffic emissions are the only source of particulate matter, for which also
number emissions have been regulated. The new diesel vehicles under EURO VI -technology
are limited not to have higher number emissions than $6*10^{11}$/km for passenger cars (the same
limit should be applied also for gasoline vehicles after 2017) and $6\text{-}8*10^{11}$/kWh for heavy-
duty vehicles. However, these limits are set only for solid particles larger than 23 nm. In
practice, this means that only particles with black carbon core are taken into account, since
the secondary particles are not considered as solid (they evaporate when the sample is heated)
and the nucleation mode particles with a non-volatile core (Rönkkö et al., 2007; Lähde et al.,
2010) have diameters well below 23 nm after evaporation of condensed matter. Thus, the
particle number emission limits mentioned above are in principle reached already when older
diesel vehicles are equipped with Diesel Particle Filter (DPF) (Samaras et al., 2005).
In addition to the emission abatement technologies and fuel type (here in principle gasoline
vs. diesel, since the global shares of gas or ethanol fuelled vehicles are very small), the
particle number emissions from traffic were highly sensitive to fuel sulphur content (FSC).
This effect is demonstrated in Table 1, where we present the relative change in road transport
PN emissions arising from the assumption of replacing all the diesel fuel with ultra low FSC
diesel, such as demanded by legislation e.g. in EU and U.S. Table 1 shows how much the
emissions would decrease, in comparison to the actual CLE scenario, if all the consumed
diesel fuel was replaced with ultra low FSC diesel. In Europe, there are some non-EU
countries for which, in the CLE scenario, the share of higher FSC diesel remains constant
until 2030. Since the total European road traffic emissions are decreasing significantly due to
the improving emission abatement technologies, the relative share of emissions from higher
FSC diesel increases with time. The table also reveals, that the expected decrease in road
transport emissions in Australia, Africa, Southern America and Russia from 2010 to 2020 (see
Fig. 4) was caused by decreasing the FSC in diesel, whereas (according to CLE scenario) in
China, India and Asia the share of ultra low FSC diesel is either not increasing or the effect of
its increase is (partly) neglected by the increasing volume of road transport.





### 3.2.5. Other significant sources

Agriculture has a significant share on particle number emissions in Russia, India and Africa and these emissions were entirely (>99 %) caused by agricultural waste burning (in which slash and burn of forests or other vegetation and forest fires were not included).

In Russia, Europe and Northern America the non-road transport emissions formed a considerable part of the emissions. However, this large non-road transport share was partly due to including the gas pipeline compressor emissions in this sector. These were dominant in Russian non-road transport emissions (95%) and constituted a major source also in Northern America (35 %). In Europe the non-road transport emissions came mainly from maritime vessels and the inland waterway transport was also a significant contributor to Northern-American emissions.

One PN source, which might have a notable share in regional emissions but was not included in this study due to lack of data on number emission factors, are brick kilns. Brick kilns are a significant source of PM especially in India and other Southeast Asia (Bhat et al, 2014).

### 3.3. Spatial distribution of emissions

Aerosol particles are short-lived climate forcers with lifetimes roughly up to a week and the aerosol number size distributions evolve rapidly especially under high concentrations close to the sources. Thus, the regional particle concentrations leading to health and climate effects cannot be defined with emissions described in country or region level, but it is essential to assess the emissions with higher spatial resolution. The gridding of emissions down to $0.5°×0.5°$ resolution, as applied in the GAINS emission model allows for estimating the regional concentrations when combined with air quality or climate models.

In the upper panel of Figure 5 the gridded global emissions are presented for the year 2010. The emissions ranged in a span of up to six orders of magnitude (note the logarithmic colour axis), from $10^{19}$ to $10^{25}$ particles per year per grid cell. The highest emissions were seen in North-Eastern China, but all the continents had various grid cells with emissions higher than $10^{24}$ #/year.

In the lower panel of Fig. 5, we have depicted the estimated change in total aerosol particle number emissions from 2010 to 2030 based on the Current legislation scenario. The main areas of significant decrease in emissions were Western Europe, Eastern United States, Brazil,



Australia, Japan and China, whereas the emissions in Africa, India and European part of
Russia were predicted to increase notably.

## 3.4.    Emission number size distributions

The number size distributions of the major source sectors is presented for years 2010 and
2030 in Fig. 6 (upper panels), respectively. Here we divided the emissions to different sectors
(e.g. according to the used fuel) than in previous figures in order to present the differences in
size distributions and total emissions related to the different fuels. Especially the domestic
combustion of coal and biomass resulted in notably different size distribution with peak
values in 20-40 nm and ~100 nm, respectively. The most significant single particle number
sources mentioned in Sect. 3.2 (road transport with diesel fuel and coke production) had peak
values in sizes from 30 to 50 nm in diameter. The difference in size distributions from
different sources was visible also when assessing the regional emissions (Fig. 6, bottom
panels). In 2010, the emissions in Africa and India were dominated with biofuel combustion
and agricultural waste burning peaking at diameters close to 100 nm, whereas the other
regions showed highest emissions around 40 nm diameter. However, the estimated increases
in Indian power production, industrial and road traffic emissions towards 2030 moved the size
distribution to smaller diameters. On the contrary, the notable decrease in Australian road
traffic emissions shifted the size distribution to larger sizes, because one of the main sources
in 2030 was estimated to be agricultural waste burning.

## 3.4.1.   Black carbon emission size distribution

The size distributions of black carbon containing particles as well as the size distribution of
the black carbon cores for year 2010, calculated with Eqs. (4-5), are presented in Figure 7.
The global black carbon mode particle emissions were dominated with diesel-fuel road
transportation, but the contributions of domestic biomass combustion and agricultural waste
burning were much higher than for the total particle numbers (compare to Fig. 6, upper left
panel). The black carbon mode count median diameter varied from 70 to 100 nm. This
variation seems to be at least partly due to the amount of vapours condensed on the black
carbon cores: the black carbon core size distributions shown in middle and right panels of Fig.
7 show more similar count median diameters of roughly 60 nm for all other sources than
industrial combustion and domestic coal combustion. The difference between the assumptions





of the composition of the coating of BC cores, i.e. the choice between coating including only
OC and coating including all $PM_1$ but BC, was significant only in industrial combustion
emissions, for which the BC core mode shifted to much smaller sizes (from ~100 nm to 30-40
nm) when assuming all $PM_1$ is condensed on BC cores. This is because in industrial $PM_1$
combustion emissions the shares of OC and BC are relatively small.
**3.5.    Future trends of emissions in different PN and PM metrics**
The projected future trends of PN emissions (UFP and FP separately) and, for comparison,
the mass emissions $PM_1$, $PM_{2.5}$ and $PM_{BC}$ are depicted in Figure 8 with indicated global
contributions of different source sectors. The significant contribution of road traffic to PN
emissions caused a decrease from 2010 to 2020 in PN emissions in both UFP and FP size
range and the decrease in UFP emissions was enhanced by the decrease in coke production
emissions. The decrease in PN emissions was predicted to stop after 2020 due to increase in
industrial emissions. This was estimated to cause a slight increase in UFP emissions from
2020 to 2030, but the global FP number emissions seemed to remain constant. Comparison to
PM mass emissions revealed that the trends of particle numbers and mass can be very
different. The major source in all the depicted mass emissions, $PM_1$, $PM_{2.5}$ and $PM_{BC}$, was
residential combustion, but $PM_1$ and BC emissions from residential combustion emissions
were estimated to decrease more than $PM_{2.5}$. As the $PM_{2.5}$ emissions showed the steepest
increase in industrial emissions, whereas the BC emissions are affected very little by
industrial process emissions, the total $PM_{2.5}$ emissions showed increase, $PM_1$ remained rather
constant and BC emissions showed clear decrease.
In most parts of the world, the future changes in UFP and FP emissions are predicted to be
rather similar (Fig. 9), but the relative change in UFP emissions is typically a bit more
pronounced than that of FP particles. However, especially in India the UFP emissions are
estimated to increase much more than FP emissions. This is because the emissions from
residential combustion and agricultural waste burning, which emit both FP and UFP, are not
increasing in India, but the industrial, traffic and coke production emissions, all emitting
mainly in UFP size range, are predicted to increase significantly (see Fig. 4). Also in Russia,
which is the other area where the number emissions are clearly increasing, the relative
increase of UFP emissions is larger than that of FP emissions. In Russia the road traffic
emissions are predicted to decrease and the increase in UFP emissions is mainly caused by



increases in emissions from industrial processes, coke production and gas pipeline
compressors. The mass emissions are depicted also in Fig. 9 for reference, but the reasons for
different regional trends are not discussed here.
**4.    Uncertainties related to the particle number emission factors**
This article has it main focus on describing the implementation of particle number emission
factors in the global GAINS emission scenario model. We present the initial results and
demonstrate the future needs for improving the emission factor database. The uncertainties in
the particle number emission factors are large and often based on gap-filling. Based on the
presented results, further research can be planned and we see these estimates, albeit uncertain,
as progress and part of the results.
The uncertainties in the emission factors are due to the following main reasons, *i*) the lack of
reliably reported measurements for the particle number emission factors and the related size
distributions, *ii*) geographic unrepresentativeness of the applied emission factors, *iii*)
application of number emissions factors based on PM mass emission factors (instead of
applying a direct number emission factor), and *iv*) a lack of representative measurements for
fuels with high and/or varying sulphur contents. High sulphur contents give rise to high
emission of particles of a very small size (<10nm), these numbers can be expected to
dominate total PN emissions in many sources.
The above listed causes for uncertainties are in many cases linked, e.g. the reason for applying
PM-based emission factors for determining number emission factors is due to the lack of
available direct number emission factors. They also make the geographic variation of
uncertainties very prominent. In Europe and Northern America, the overall uncertainties can
be estimated to be relatively small, both in terms of current and future emissions. This is
because most of the emission factor measurements have been conducted in these continents
and in both the dominant sources of emissions are road traffic and residential wood
combustion, both with well-established direct number emission factor database for different
emission abatement technologies. On the contrary, the emission factors for the dominant
particle number sources in Asia (including China, India and Russia) are in most cases based
on only few (often European or American) studies, and the effect of emission abatement
technologies is typically based on $PM_1$ emission factors. Also the pronounced wood





combustion emissions from cooking in Africa are based on emission factors from (traditional
western) heating stoves and are thus rather uncertain.
In the following we discuss shortly the most important individual causes for uncertainties in
the results presented in Sect. 3.
o      Applying PM-based emission factors in general
The emission abatement technologies have typically different removal efficiencies for
particles with different diameters. However, when the emission factors for different
technologies are determined by simply scaling the emission factor with the corresponding
change in PM emission factor, the PSD remains unchanged. This may result in erroneous
estimates of $EF_{PN}$, e.g. if a source with high emissions of fine particles and condensable
vapours is controlled with a removal technology for the fine particles, the formation of
ultrafine particles from the vapours may increase due to drastic decrease in the condensation
sink for the vapours and coagulation sink for the freshly formed particles.
o      Effect of sulphur on PSDs and emission factors
It is well known that sulphuric acid, formed from $SO_2$ after oxidation to $SO_3$, is a key player
in atmospheric new particle formation. It has been also shown in many studies that, by
increasing the fuel sulphur content, the emissions of ultrafine particles are increased (e.g.
Rönkkö et al., 2013).  However, the nucleation mode particles formed from sulphur (and
other condensable vapours) are often not well, in some cases even at all, represented in the PN
emission factors and PSDs in the literature. Some instruments applied for the measurements
are not able to measure concentrations of particles with diameters below 10 nm, and in some
cases the nucleation mode particles are evaporated before they are detected. It can be
expected that, by making new experiments on the PN emission factors and PSDs with
instruments suitable for detection of nucleation mode particles, the overall figure of UFP
emissions will alter significantly. It might be also possible to derive semi-empirical estimates
of the nucleation mode particle emissions by taking into account the $SO_2$ emissions.
Additional uncertainties related to sulphur emissions arises from the lack of emission factors
for different fuel sulphur contents in sources other than road traffic. Especially in coal
combustion the emissions can be expected to depend heavily on the coal sulphur content.





Also for the road traffic emissions, the uncertainties are considerably higher for higher FSC
diesel than for ultra low FSC diesel or gasoline.
o    Coke production
Emission factor for coke production is based on $PM_1$ emission factors and the conversion
from mass to number factor and the particle number size distribution are derived from a
publication by Weitkamp et al. (2005), in which the authors study the emissions from a large
coke production facility near Pittsburgh, U.S. Other studies for comparing the number size
distribution related to coke production, especially in Asia, are needed for verifying the drastic
impacts of coke production to regional aerosol emissions. Furthermore, the effects of
emission abatement technologies – such as cyclone, 1- and 2-field electrostatic precipitators
and high efficiency dedusters – on the particle size distribution and number emission factor
need to be studied.
o    Residential coal combustion
Residential coal combustion number emission factors are PM-based and were produced with
particle size distributions taken from Bond et al. (2002). Further studies for different coal
types, including varying sulphur contents, and stove technologies are needed to better
estimate the share of residential coal combustion on the particle number emissions especially
in China.
o    Residential wood combustion in traditional cooking stoves
The emission factors for the cooking stoves e.g. in African and Asian countries have been
adapted from no-control emission factors for heating stoves, which are mostly based on
Northern-European and –American studies. Obtaining emission factors for traditional cooking
stoves down to a three stone fire, would give better picture on the residential combustion
emissions especially in Africa. Furthermore, estimating the dung combustion activity levels in
countries other than India could alter the overall figure to some extent.
o    Power plant and industry emissions





The emission factors for power plants and industry are all PM-based, which causes
uncertainties especially when assessing the future emissions with improved technologies.
Also the fuel sulphur contents are not taken into account, which increases the uncertainty
levels.
o       Effects of ambient conditions on emissions
The numbers and size distributions of emitted particles depend also on the ambient conditions
in which they are emitted. The volatility of vapours is strongly dependent on temperature,
which naturally causes evaporation when fuel is heated. Some of the vapours that do not
effectively condense onto particles and/or form new nucleation mode particles in room
temperatures may still be condensable when temperature is lower. This would affect the
emissions most probably in the colder parts of the world and especially in winter.
**5.     On the effects of anthropogenic emissions on particle number**
**concentrations**
In this paper we have presented the first results of global anthropogenic particle number
emissions from the GAINS model. It is important to note that e.g. the future trends presented
here should not be interpreted as trends for future particle number concentrations, because the
relation between particle number emissions and number concentrations are far from linear.
Typically, particle number concentrations vary much less than the emissions, because *i*) in the
areas of low anthropogenic emissions the natural emissions and aerosol formation play a
relatively more important role (Paasonen et al., 2013a) and *ii*) the most efficient sink for the
aerosol particles is their coagulation with larger particles (e.g. Kerminen et al., 2001).
Because this coagulation sink of particles correlates in many cases with the number emissions
(e.g. in the street canyons both the number concentrations and sink are high, and in general
both increase when approaching the emission source), the implementation of the GAINS
number emissions to air quality or climate models even with the higher spatial resolution
(0.5°×0.5°) may lead to overestimating the concentrations. In order to better approach e.g. the
health effects of particle number concentration within cities, it is possible to downscale the
GAINS emissions to a street canyon scale with the methods presented by Kiesewetter et al.
31  (2014).



Comparison of the global emission trends of different aerosol concentration metrics (Figs. 8-
9) reveals their different predicted trends. The emissions of black carbon aerosol, the main
aerosol component causing global warming, are predicted to decrease in the future, whereas
the emissions of cooling aerosols, i.e. mass emissions excluding BC (cooling due to scattering
of solar radiation) and the number emissions of FP (acting as cloud condensation nuclei,
CCN) are predicted to increase or remain quite constant. However, it should be noted that the
climate effects do not follow directly the emissions, especially in the case of cloud droplet
formation. There are several processes, which can either overrule or dampen the formation of
cloud droplets from emitted FP. Firstly, the UFP from both anthropogenic emissions and
atmospheric new particle formation grow to CCN-sizes, and this growth produces often much
more CCN than primary FP emissions, and secondly, the boundary layer height and dilution
also affect the concentration levels resulting from the emissions (Paasonen et al., 2013a).
Thirdly, the cloud droplet concentration (at least partly) saturates when CCN concentrations
increase, which lessens the cloud forming effect of FP emitted in moderately or more polluted
areas (e.g. Gultepe and Isaac, 1999).
**6.    Conclusions**
The aerosol particle number (PN) emission factors and the related size distributions have been
implemented in the global GAINS model. The regional PN emissions are dominated by
different sources than e.g. the particle mass emissions. In most parts of the world the
emissions from diesel fuelled road vehicles were the major source in 2010. Other significant
sources for particle numbers were residential combustion of biofuels and coal (especially in
China, India and Africa), coke production (Russia and China), industrial combustion and
processes (Russia, China and India) and gas pipeline compressors in Russia. However, the PN
emission factors for residential coal combustion, coke production and gas pipeline
compressors have high uncertainties, which can be reduced only with further new
experimental studies on the emission factors.
According to the current legislation scenario, the PN emissions are expected to decrease
significantly by 2030 in Europe, Northern and Southern Americas and Australia (64%, 49%,
26% and 76%, respectively), mainly because of introduction of Diesel Particulate Filters
(DPF) in order to comply with new diesel vehicle legislation; the DPFs cut efficiently both
particle mass and number emissions the transport emissions. In Southern-America and



Australia the decrease in road traffic emissions is also partly due to intended switch to ultra-
low sulphur content fuels, which are already the only fuel type in use in Northern-America
and most of the European countries. Also in China the total PN emissions are estimated to
decrease by 23% from 2010 to 2030, mainly due to the decreases in coke production and
residential coal combustion emissions. However, in India the emissions are increasing by over
80 % from 2010 to 2030, in Russia by 37% and in the rest of Asia by 19%, whereas in Africa
the emissions are estimated to increase only by 7%.
The number size distributions of particles differ significantly depending on the source. In
terms of the major number sources, traffic, coke production and residential coal combustion
show highest emissions in ultrafine particle (UFP) size range, with diameters between 30 and
50 nm, whereas the residential biofuel combustion and agricultural waste burning, as well as
industrial combustion, show peaks with diameters around 100 nm. These differences,
naturally, cause variation in the total number size distributions of emitted particles in different
parts of the world.
The sizes of emitted particles are important when assessing the impacts of the emitted
particles. The globally significant climate impact of particle number concentrations arises
from the aerosol-cloud interactions, i.e. the activation of particles with diameters close to or
over 100 nm as cloud droplets. On the other hand, the adverse health effects related to particle
number concentration are coupled with UFP concentrations. This, together with the
dominance of traffic emissions in this size range and the fact that road traffic is a pollution
source very close to our every day life, arises need for better assessment of size segregated
PN emissions also in the population health perspective. Thus, this work provides input for
both climate and air quality modelling and makes the evaluation between the effects of the
future changes in aerosol number emissions and aerosol mass emissions possible.
However, the work described in this paper is the first implementation of the particle number
emissions to an emission scenario model such as GAINS. In order to improve the estimates of
current and future PN emissions, more experiments on the PN emission factors and size
distributions of the sources indicated in Sect. 4 are crucial, as well as a thorough reassessment
of the effects of fuel sulphur content and ambient conditions on the emission.
**Acknowledgements**





This work was funded by the Academy of Finland Centre of Excellence (grants no. 1118615
and 272041), European Commission 7th Framework projects ECLIPSE (Project no. 282688),
PEGASOS (265148), TRANSPHORM (243406) and 'Assessment of hemispheric air pollution
on EU air policy' (contract no. 07.0307/2011/605671/SER/C3) and by the Nordic Top-level
Research Initiative (TRI) Cryosphere-Atmosphere Interactions in a Changing Arctic Climate
(CRAICC). We thank Leonidas Ntziachristos and Ilias Vouitsis at Aristotle University of
Thessaloniki (Greece) for help and assistance in applying the emission factors for road
transport sector and Professor Qiang Zhang from Tsinghua University (Beijing, China) for the
spatial distribution of Chinese power plants for 2000, 2005, and 2010.

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



Table 1. The relative difference in annual road transport PN emissions between CLE scenario
with an additional assumption that all diesel fuel (consumed in road transport) has ultra low
sulphur content (FSC = 10 ppm) and the actual CLE scenario. The lowest row shows the
change in total emissions from all sources. Note that e.g. in Europe, the increasing effect is
due to a combination of drastically decreasing emissions in most of countries and a small
remaining share of high FSC fuel in some countries, increasing thus the proportion of the high
FSC contribution to total emissions.

|                        | 2010  | 2020  | 2030  |
|------------------------|-------|-------|-------|
| Europe                 | -5%   | -8%   | -24%  |
| N. America             | 0%    | 0%    | 0%    |
| Russia                 | -48%  | -1%   | -3%   |
| China                  | -30%  | -31%  | -33%  |
| India                  | -32%  | -41%  | -39%  |
| Asia                   | -44%  | -29%  | -32%  |
| S. America             | -35%  | -1%   | -2%   |
| Africa                 | -55%  | -7%   | -8%   |
| Australia              | -51%  | 0%    | 0%    |
| **Global road traffic** | -32%  | -21%  | -27%  |
| Global total           | **-11%** | **-5%** | **-6%** |



**Figures and figure captions**

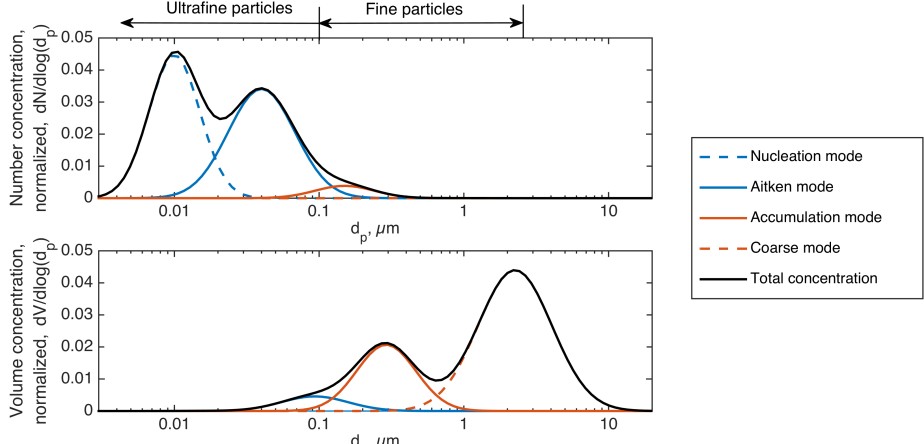

Figure 1. Number size distribution of a fictional and simplified particle population within the
planetary boundary layer with four lognormal particle size modes (upper panel) and the same
population represented with mass size distribution (lower panel). Note that in literature it is
common to use term "fine particles" (FP) when referring to all particles with diameters below
2.5 µm, including ultrafine particle (UFP) size range. However, in this article we exclude
UFP size range from FP.





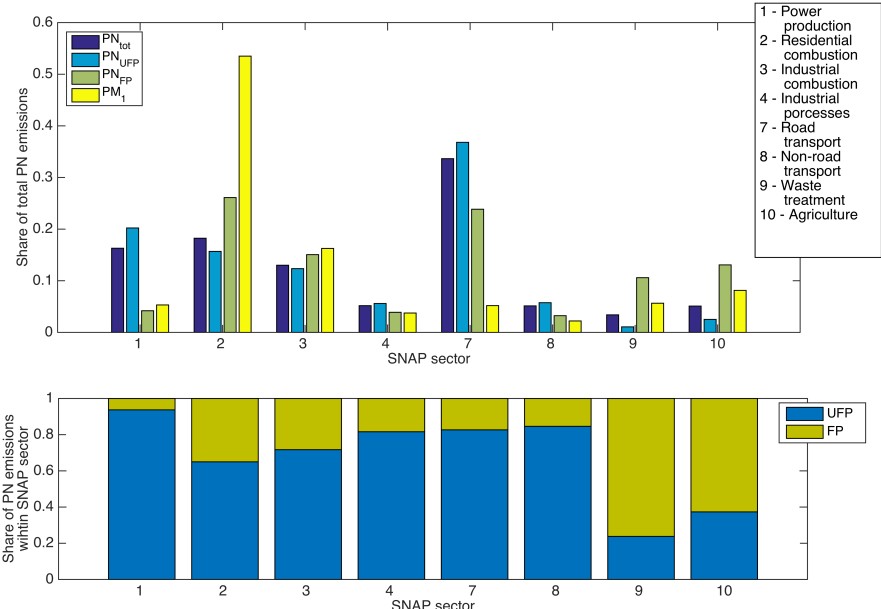

Figure 2. Upper panel: shares of different source sectors in number emissions of all (PN$_{tot}$),
ultrafine (PN$_{UFP}$) and fine (PN$_{FP}$) particles and in aerosol mass emissions of particles with
diameters below 1 µm (PM) for 2010. Lower panel: shares of UFP and FP in PN emissions
for each SNAP-sector.

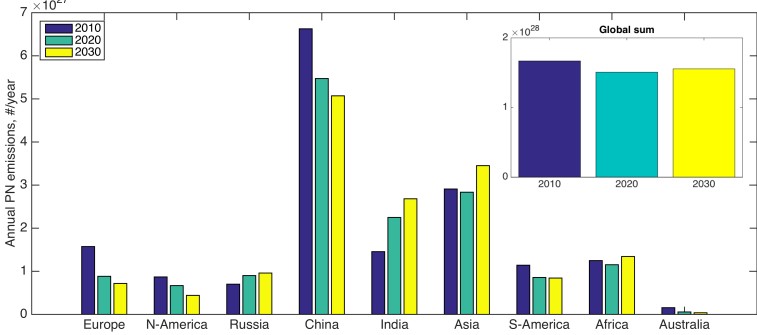

Figure 3. Evolution of continental anthropogenic particle number emissions from 2010 to
2030 according to the current legislation scenario in different parts of the world and the whole
world.





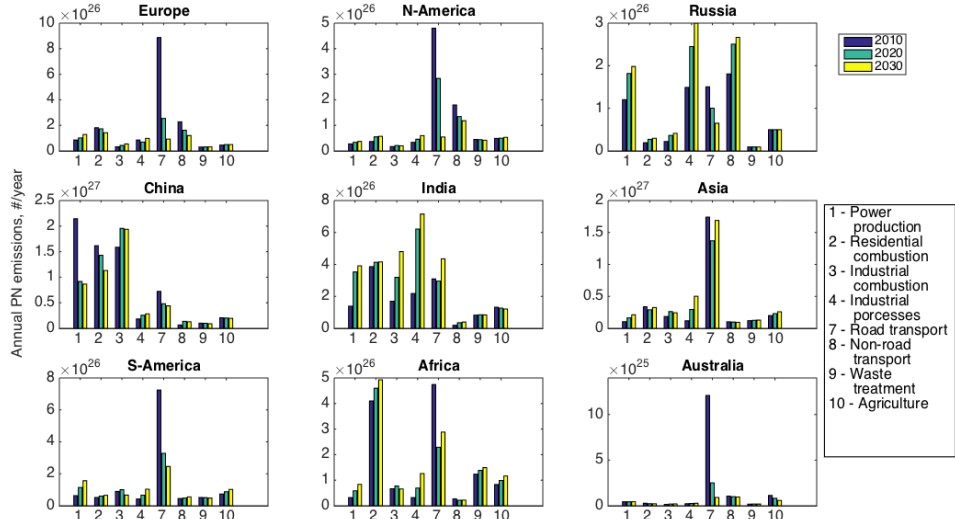

2    Figure 4. Contributions of different source sectors to particle number emissions in different

3    parts of the world, from 2010 to 2030. Note the different Y-axis scales.





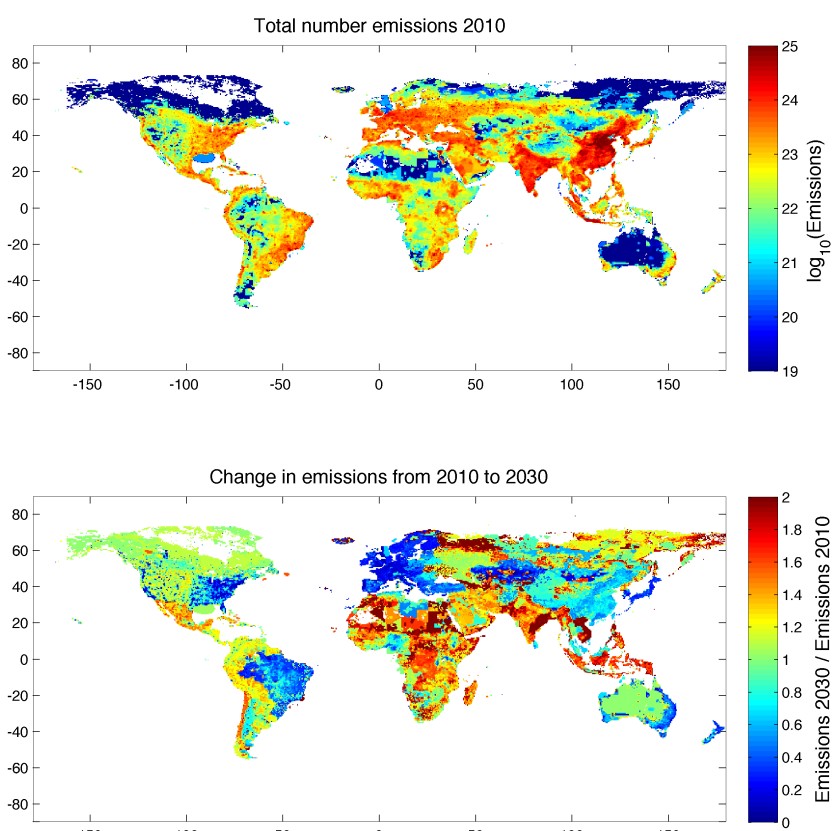

2   Figure 5. Spatial distribution of global continental anthropogenic particle number emissions

3   in units #/grid box, with grid box size 0.5°×0.5° (upper panel) and predicted relative change

4   in particle number emission from 2010 to 2030 (lower panel).





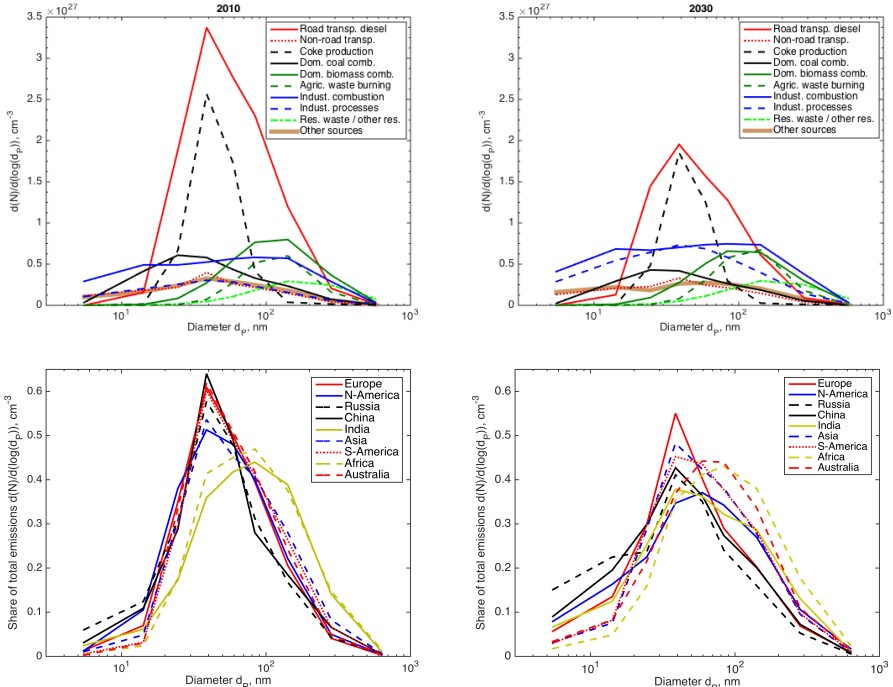

Figure 6. Particle number size distributions of the major global aerosol emission sources
(upper panel) and normalized number size distributions for each region (lower panel). The left
side figures are for 2010 and the right side ones for 2030.

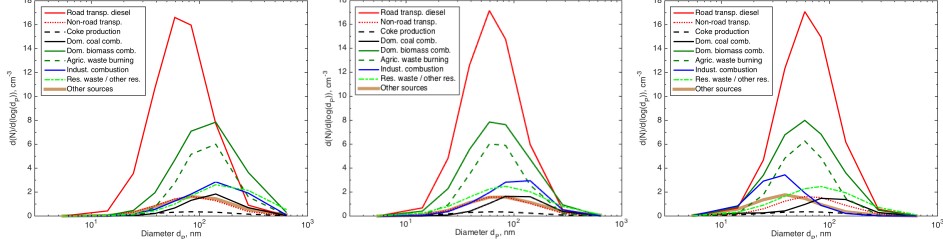

Figure 7. Estimated global number size distributions of the black carbon mode particles (left
panel) and of their black carbon cores, assuming only OC is condensing on the BC cores
(middle panel) and assuming all $PM_1$, except BC, is has been formed by condensing on the
BC cores (right panel). The source categories are the same as in Fig. 6.



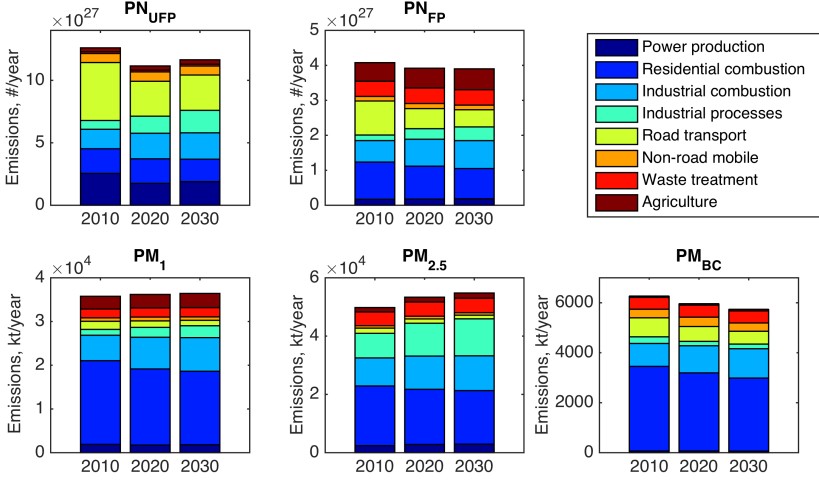

Figure 8. Shares of different source sectors to the future global trends particle number and mass emissions under current legislation scenario: PN emissions in ultrafine and fine size ranges and particle mass emissions $PM_1$, $PM_{2.5}$ and black carbon.

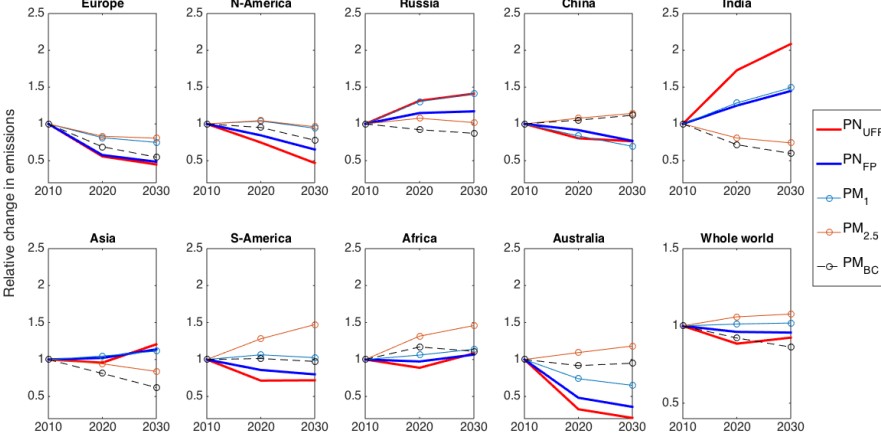

Figure 9. Continental future trends of particle number and mass emissions under current legislation scenario. Emissions are normalized to unity in 2010. Note the different y-axis scale in the subplot for the Whole world.