# Peer review of "Continental anthropogenic primary particle number"

_Atmospheric Chemistry and Physics, 2015_

## Referee Comment (RC1) · Anonymous Referee #1 · 21 Feb 2016

The authors present an implementation of global particle number emission factors to the GAINS model. Such work is very important for atmospheric aerosol studies, to move more towards a complete description of the aerosol also from the emission side. The manuscript gives an overview of the applied methodology, and also gives a general description of the results of the application. The main sources of uncertainty are also discussed in some detail. The manuscript is an valuable addition especially for modellers in the atmospheric community, and it should be published. There are some comments that should be answered before publication, though.

Generally, the manuscript (according to the title) discusses primary particles. As also discussed in parts in the text, some sources emit vapours that in some conditions quickly form particles. For some sources, this is not captured by the emission factors, because such particles are purposefully removed e.g. by heating, while for some

sources these particles will be present. This is an issue for example for the sulphur-related emissions. Is there a distinction between the "actual-primary" and "quickly-formed secondary" particles done in the current work? Are there possible errors in the emission estimates due to this?

Point-by point comments:

p4, line 3: the authors refer a couple of times to coagulation as the main sink of particle number. The study that they are referring to (Kerminen et al., 2001) refers to recently-born particles, and should not be generalized to all sub-100 nm particles. For example, particles with a diameter of 50 nm have ca. 50-fold lifetime due to coagulation when compared to 5-nm particles, and other loss mechanisms will become important as particles grow. Please change the text to reflect this. Coagulation will of course be important immediately after emission and at high concentrations, and the effect of increasing number with decreasing mass emission is thus possible.

p 7 line 1: it would be nice to know the number and spacing of the size sections, especially also the lower limit of the size sections.

P7 l8: uncertain language: this statement seems to me a bit vague. Measurement methods are generally less uncertain in these size ranges. As explained later, authors mean that even a small error in the number of near- 1um particles causes a large error in the number of smaller particles when converting mass concentrations. This is a valid and good point, but the wording here should be fixed.

P7 l18 Which TNO study is meant? Two references were originally given.

p12 l9-13: What is meant here? Surely the emissions are dependent on the sulphur removal techniques as well as the fuel content, but this has not been accounted for in the modelling? Or is something else meant here? These sentences should be clarified, as well as the the reasoning for this factor only causing an underestimation.

p13 l4: please also give a pointer to the specific reason for the inconsistency discussed

in Sect 4.

p14 line 31: neglected is probably not the right word here?

page 15, section 3.3. and Fig 5: I think using the measure of particles/grid cell is not the right choice, especially if this is then used for comparisons between different geographical areas. This because the grid area is different near the equator than near the poles (the gridding is based on latitudes and longitudes), and for comparison purposes the emissions should be normalized by the grid cell area.

p18, l23: "In Europe and Northern America, the overall uncertainties can be estimated to be relatively small, both in terms of current and future emissions." This is a quite optimistic sentence. The word 'relatively' makes it probably correct in terms of comparisons with other estimates, but there remain several factors in measurement technologies, changes in abatement technologies, ultrafine particle concentration variability etc. that I think that a statement that uncertainties are small is yet preliminary.

p19 l10 -> "This may result ..." this is an interesting point. How does the GAINS model see particles that are formed due to a decrease CS from the original source? Such particles could also be considered secondary particles, should such particles be covered by emission inventories? See also general comments.

p20 line 6, "Effects of ambient conditions on emissions": This is an important point. Here, the authors consider only a very limited temperature effect on the emissions; actually, also many other factors, such as the existing aerosol concentration, might affect the emission, especially if some secondary particles are also considered in the emission factors. This could be elaborated on in this section.

p22 l32: remove "the transport emissions". Also, remove hyphen from "Southern America"

Table 1, l4 (page 33) : This sentence is very complex: what is decreasing or increasing, and what causes what? How can high FSC cause effects when it is assumed to be

replaced by low FSC? This is explained better in the text, but understanding the table is hard with this caption.

---

## Referee Comment (RC2) · Anonymous Referee #2 · 7 Mar 2016

The authors add primary aerosol particle number distributions to an emissions scenario model (GAINS). Number emissions are presented for a group of components (e.g. sulphate, OC) and explicitly for black carbon (BC). The primary emissions of number are based on a three-mode representation of the aerosol smaller than 1 um diameter. The mass emissions already in the model are distributed according to various observations in the literature for primary particle emissions. The focus is on the year 2010 and those estimates are compared with estimate for 2020 and 2030 based on future scenarios. Overall, I believe it is a very useful contribution to the literature and appropriate to ACP. The paper is generally well presented, but there are several places where some improvements are needed. I have several minor comments.

Specific comments:

1) Page 1, line 16 - Define "GAINS".

2) Page 2, line 12 – Avoid "Clearly" (and use of similar terms in other places). Arctic Haze is one example of a broad aerosol that has more numbers >100 nm than smaller. Biomass burning is another aerosol that provides more larger particles. Most often, UFP dominate the numbers, but not always.

3) Page 2, lines 16-19 – In fact, correlations of particle volume (and sulphate mass concentration) and number have been published (e.g. Leaitch et al., Tellus, 1986; Hegg and Russell, JGR, 2000), and these scenarios may well exist as often as not.

4) Page 3, lines 1-3 – Explain what you mean by neglecting the warming from BC due to their ability to form cloud droplets. This is an important point, but it is poorly described here.

5) Page 3, lines 3-5 – Discussions of the impact of reducing sulphur emissions started at least as far back as 1989 (Wigley, Nature).

6) Page 3, lines 5-8 – Please re-write this sentence so the meaning is clear.

7) Page 3, line 10 – New particle formation can result from primary emissions. Just refer to this as particle nucleation in the atmosphere.

8) Page 3, Lines 12-15 – Another sentence needing clarification. I believe you intend this as an indication of particle formation during direct emissions, but it follows the sentence about nucleation in the atmosphere without connecting with direct emissions.

9) Page 3, line 32 – Comment 3 above relates to this also.

10) Both Amann references on pages 4 and 5 – The first refers to EMS and the second to TSAP. Both of these need explanations.

11) Page 6, line 26 – define TNO.

12) Page 6, line 28 to page 7, line 3 - You refer to sizes and sectors using "i", which is

confusing. In equation 1, "i" is used to refer to a region, whereas in equation 2 it refers to a size class. Make it easier.

13) Page 6, Line 27 – Add a sentence or two to indicate what factors are used to distinguish primary emissions without influence from secondary processes.

14) Page 7, lines 4-6 - EFpn:S? Is it intended as a plural of EFpn?

15) Page 8, line 23 – "New PSDs were" or "A new PSD was"?

16) Page 9, lines 3-5 - By black carbon mode, I assume you mean "pure" black carbon mode. Please clarify in the text.

17) Page 9, lines 9-14 - The Sorensen et al. work focussed on the morphology of larger particles. Their techniques and interest limited the discussion of smaller particles, and indeed their micrographs indicate BC particles smaller than 50 nm. See Liggio et al. (Environ. Sci. Technol. 2012, 46, 4819−4828). It may be that your modal representation includes sufficient BC at sizes smaller than 50 nm, but the discussion needs to be a little more objective on this point.

18) Page 9, lines 18-21 – There has been significant debate as to what constitutes primary OC. This should be reflected in the response to comment 13 above.

19) Page 9, line 30 – "except for BC formed through condensation". This is confusing. I assume you mean PM1 minus the BC component. Please clarify.

20) Page 18, lines 23-24 – "are smaller"

21) Page 19, line 19 – ". . . the primary emissions of . . ."

22) Page 21, line 17 – "primary emissions"

23) Page 21, line 21 – what do you mean by aerosol formation: secondary processes; nucleation?

24) Page 22, lines 1-15 – I am surprised that you have chosen to focus only on the

warming aspect of BC, when your work seems suited to address the question about how much BC affects the number distributions and therefore the number concentrations of CCN; see Chen et al 2010, GRL. Estimating how the total number concentrations compare with the BC number concentrations from Figures 6 and 7 is very difficult, and in Figures 8 and 9 you only refer to BC mass. I would like to see a plot comparing the average number distribution from Figure 6 and the average BC mode and core number distributions from Figure 7. Of course there are still particles formed from secondary processes to be considered, but more knowledge of the importance of BC to primary emissions would be immensely useful.

---

## Author Comment (AC1) · 25 Apr 2016

**The authors' response to the comments by Referee 1**

The referee's comments are presented in *italic* font, followed with our responses

We thank the referee for a thorough and constructive review, which clearly improves our manuscript. Below we reply to the comments by the referee and describe how they are captured in the manuscript.

*The authors present an implementation of global particle number emission factors to the GAINS model. Such work is very important for atmospheric aerosol studies, to move more towards a complete description of the aerosol also from the emission side. The manuscript gives an overview of the applied methodology, and also gives a general description of the results of the application. The main sources of uncertainty are also discussed in some detail. The manuscript is an valuable addition especially for modellers in the atmospheric community, and it should be published. There are some comments that should be answered before publication, though.*

*Generally, the manuscript (according to the title) discusses primary particles. As also discussed in parts in the text, some sources emit vapours that in some conditions quickly form particles. For some sources, this is not captured by the emission factors, because such particles are purposefully removed e.g. by heating, while for some sources these particles will be present. This is an issue for example for the sulphur-related emissions. Is there a distinction between the "actual-primary" and "quickly-formed secondary" particles done in the current work? Are there possible errors in the emission estimates due to this?*

In this work we apply reported particle number size distribution measurements within fresh plume, instead of stack measurements, for deriving the most part of the emission factors. Thus, the quickly formed secondary particles are included in the emissions factors, as far as the measurement methods have allowed their detection. However, with this basis for emission factors, we are not able to directly make distinction between the "actual-primary" and "quickly-formed secondary" particles as suggested by the referee (clarified in the revised manuscript on page 3, lines 16-18, and page 7, lines 5-10). Our calculations of black carbon containing particles would, in principle, allow a rough kind of distinction towards what the referee suggests, if all the particles smaller than the assumed black carbon mode were taken as "quickly-formed secondary" particles. We decided not to do this in this manuscript, partly due to the other remark made by the referee: there are differences in the measurement techniques (in terms of how and if the non-solid particles are detected) used in the literature applied for the determination of the GAINS emission factors. We find that the deeper analysis of the related uncertainties and the revision of the emission factors with respect to the emissions of these non-solid, e.g. sulphur-related, particles requires further work and we highlight the importance of pursuing this in the near future.

The possible errors arising from partial inconsistency treatment of quickly formed

secondary particles between different source sectors are discussed in Sect. 4 under title "Effect of sulphur on PSDs and emission factors". By lifting these uncertainties as the second of the three general sources of uncertainties in Sect. 4.1, we point out the importance of this issue and hope it will allow for future funding on this topic. We modified Sect. 4 in terms of dividing the sources of uncertainties under topics of "General" and "Sector-specific" causes for uncertainties, Sub-sections 4.1 and 4.2, respectively (pages 20-22). With this, we moved the paragraph "Effects of ambient conditions on emissions" to the end of sub-section 4.1 (more on the changes in this paragraph, please see the answer for the specific question below).

*Point-by point comments:*

*p4, line 3: the authors refer a couple of times to coagulation as the main sink of particle number. The study that they are referring to (Kerminen et al., 2001) refers to recently-born particles, and should not be generalized to all sub-100 nm particles. For example, particles with a diameter of 50 nm have ca. 50-fold lifetime due to coagulation when compared to 5-nm particles, and other loss mechanisms will become important as particles grow. Please change the text to reflect this. Coagulation will of course be important immediately after emission and at high concentrations, and the effect of increasing number with decreasing mass emission is thus possible.*

We have made the following (underlined) additions to the text according to this comment:

P4, line 8: "Additionally, because the main removal mechanism of the smallest of UFP in the atmosphere is their coagulation to larger particles (e.g. Kerminen et al., 2001),…"

P23, lines 3-4: "…the most efficient sink for the smallest of aerosol particles in nucleation mode is their coagulation with larger particles…"

*p 7 line 1: it would be nice to know the number and spacing of the size sections, especially also the lower limit of the size sections.*

The information on the size sections was unintentionally left out from the manuscript. We have added a table (Table 1) showing the size ranges of the size classes and made the following additions to the text:

P7, line 16-17: "The diameter ranges for the size classes applied for the GAINS emissions are shown in Table 1."

P8, line 7: "(see Table 1)"

*P7 l8: uncertain language: this statement seems to me a bit vague. Measurement methods are generally less uncertain in these size ranges. As explained later, authors mean that even a small error in the number of near- 1um particles causes a large error in the number of smaller particles when converting mass concentrations. This is a valid and*

*good point, but the wording here should be fixed.*

The indicated sentence has been removed, and the following sentence has been modified as:

P7, line 22-27: "However, deriving an $EF_{PN}$ directly from the $EF_{PM1}$ would make the $EF_{PN}$ very sensitive to the estimated number of close to 1 μm particles, since their mass is significantly larger in comparison to the mass of those with diameter below or around 100 nm. Thus, emission factors for PM in the size range 10-300nm ($EF_{PM0.3}$) were first derived from $EF_{PM1}$ based on literature on emission mass size distributions and particle densities (M. Kulmala et al., 2011, H. Denier van der Gon et al., 2010)."

*P7 l18 Which TNO study is meant? Two references were originally given.*

"Study" is now replaced with plural and references are repeated (p7, line 31).

*p12 l9-13: What is meant here? Surely the emissions are dependent on the sulphur removal techniques as well as the fuel content, but this has not been accounted for in the modelling? Or is something else meant here? These sentences should be clarified, as well as the the reasoning for this factor only causing an underestimation.*

We meant that the applied emission factors are not dependent on fuel sulphur content or sulphur removal technologies, but the choosing of the words was wrong. This is now corrected (p12, line 24: "the developed emission factors for (coal-fired) power plants are not dependent…").

The emission factors for power plant emissions are based on the previous TNO studies, which have focused on European emissions. Since in Europe the coal-fired power plants are in practise all equipped with some level of sulphur removal technology, the sulphur related PN emissions are estimated to be minor. However, in countries where this is (partly) not the case, the sulphur related addition to power plant emissions can be expected to be significant or even dominant. This has now been clarified in the manuscript:

p12, lines 26-30: "The applied power plant emission factors are designed for power plants in Europe, where sulphur removal technologies are in place. This may cause significant underestimation in the emission estimates for power plants using high sulphur fuels (for other power production sources than coke production) in many parts of the world, where a significant fraction of the power plants are not equipped with such technologies."

*p13 l4: please also give a pointer to the specific reason for the inconsistency discussed in Sect 4.*

We are not sure which inconsistency the referee is pointing at, since the given line (p13 line 4 in the original manuscript) is the title of the section "Industrial combustion and processes", which ends with a pointer to Sect. 4.

However, it seems appropriate (and the referee might have meant) to add a pointer to the uncertainties related to the connection between sulphur emissions and PN emission factors in the end of the Sect 3.2.1 discussed above. We replaced the last sentence of Sect 3.2.1. with:

P13, lines 3-5: "These uncertainties, also influenced with the general uncertainties related to the representativeness of the PN emission factors for nucleation mode sulphate/sulphuric acid particles, are discussed in more detail in Sect. 4."

*p14 line 31: neglected is probably not the right word here?*

No, it is not. Replaced with "invalidated" (p15, line19).

*page 15, section 3.3. and Fig 5: I think using the measure of particles/grid cell is not the right choice, especially if this is then used for comparisons between different geographical areas. This because the grid area is different near the equator than near the poles (the gridding is based on latitudes and longitudes), and for comparison purposes the emissions should be normalized by the grid cell area.*

This is right. We have replotted the figure in units $\#/km^2$. With this we noted that there was a mistake in text saying that the emissions ranged in span from $10^{19}$ to $10^{25}$ per grid cell. This span was not the original, but the colour values for the figure were set to this (i.e. all values below $10^{19}$ were shown as $10^{19}$). We have reviewed this sentence as follows:

P16, lines 16-18: "The gridded emissions ranged in a span of various orders of magnitude (note the logarithmic colour axis in Fig. 5, where the values below $10^{16}$ particles/$km^2$/year are shown as having the value of $10^{16}$)."

*p18, l23: "In Europe and Northern America, the overall uncertainties can be estimated to be relatively small, both in terms of current and future emissions." This is a quite optimistic sentence. The word 'relatively' makes it probably correct in terms of comparisons with other estimates, but there remain several factors in measurement technologies, changes in abatement technologies, ultrafine particle concentration variability etc. that I think that a statement that uncertainties are small is yet preliminary.*

Correct. We have modified the sentence pointed out by the referee as (p19, lines 23-25): "In Europe and Northern America, the overall uncertainties, even though significant in absolute values, are smaller in comparison to the other continents, both in terms of current and future emissions."

*p19 l10 -> "This may result ..." this is an interesting point. How does the GAINS model see particles that are formed due to a decrease CS from the original source? Such particles could also be considered secondary particles, should such particles be covered by emission inventories? See also general comments.*

For the PM-based emission factors the change in condensation sink CS (from the same source) due to technological improvements doesn't affect the emissions at all, but the relative change in each size bin is directly proportional to the change in PM (mass) emission factor. For direct emission factors see the answer for the general comment.

*p20 line 6, "Effects of ambient conditions on emissions": This is an important point. Here, the authors consider only a very limited temperature effect on the emissions; actually, also many other factors, such as the existing aerosol concentration, might affect the emission, especially if some secondary particles are also considered in the emission factors. This could be elaborated on in this section.*

We have underlined that the explained temperature effect is an example (P20, line 2), and added the following to the end of this section:

P21, lines 13-18: "Also the particle concentrations prior to emission can be presumed to affect the PN number emissions (at least when the immediate formation of secondary particles are considered as PN emission), due to the competition of (emitted) vapour uptake between new particle formation and condensation to pre-existing particles. These kinds of effects are, however, issues for future research and their impact cannot be implemented directly to the GAINS model."

*p22 l32: remove "the transport emissions". Also, remove hyphen from "Southern America"*

Done.

*Table 1, l4 (page 33): This sentence is very complex: what is decreasing or increasing, and what causes what? How can high FSC cause effects when it is assumed to be replaced by low FSC? This is explained better in the text, but understanding the table is hard with this caption.*

This explanation has turned out to be extremely difficult to be put in words. We have now rewritten the caption as follows:

P35, lines 4-11: "The relative change in annual road transport PN emissions in comparison to the CLE scenario, if (in addition to the technological advancements described in CLE scenario) all the diesel fuel (consumed in road transport) is assumed to be replaced with ultra low sulphur content –fuel (FSC = 10 ppm). The lowest row shows the change in total emissions from all sources. Note that e.g. in Europe, the impact increases with time, because in the CLE scenario the emissions decrease drastically in most countries, but a small share of high FSC fuel remains present in some (non-EU) countries. Thus, the proportion of the high FSC contribution to total emissions in the CLE scenario increases with time."

---

## Author Comment (AC2) · 25 Apr 2016

**The authors' response to the comments by Referee 2**

The referee's comments are presented in *italic* font, followed with our responses

We thank the referee for a thorough and constructive review, which clearly improves our manuscript. Below we reply to the comments by the referee and describe how they are captured in the manuscript.

*The authors add primary aerosol particle number distributions to an emissions scenario model (GAINS). Number emissions are presented for a group of components (e.g. sulphate, OC) and explicitly for black carbon (BC). The primary emissions of number are based on a three-mode representation of the aerosol smaller than 1 um diameter. The mass emissions already in the model are distributed according to various observations in the literature for primary particle emissions. The focus is on the year 2010 and those estimates are compared with estimate for 2020 and 2030 based on future scenarios. Overall, I believe it is a very useful contribution to the literature and appropriate to ACP. The paper is generally well presented, but there are several places where some improvements are needed. I have several minor comments.*

*Specific comments:*

*1) Page 1, line 16 - Define "GAINS".*

Done.

*2) Page 2, line 12 – Avoid "Clearly" (and use of similar terms in other places). Arctic Haze is one example of a broad aerosol that has more numbers >100 nm than smaller. Biomass burning is another aerosol that provides more larger particles. Most often, UFP dominate the numbers, but not always.*

We agree and have made the following changes according to this suggestion by the referee:

P2, lines 12-16: Sentence modified and divided in two sentences: "Aerosol number concentrations are typically dominated by particles in ultrafine particle (UFP) size range, with $d_p < 0.1$ μm, or the smaller end, roughly $< 0.3$ μm, of fine particles (FP, here $0.1 – 2.5$ μm). On the contrary, the mass concentration depends mostly on the larger and heavier, but typically fewer FP, with $d_p > 0.1$ μm (see Fig. 1 for schematic representation). "

P11 line 7: deleted "clearly"

P11 lines 19-20: "clearly" taken out and the beginning of the sentences modified as: "In 2010, China had by far the major PN emissions with 40 % estimated share of the global emissions,…"

*3) Page 2, lines 16-19 – In fact, correlations of particle volume (and sulphate mass concentration) and number have been published (e.g. Leaitch et al., Tellus, 1986; Hegg and Russell, JGR, 2000), and these scenarios may well exist as often as not.*

We find that the articles the referee points at are not entirely relevant for this comparison. The number concentrations showing good correlations with aerosol sulphate mass in Leaitch et al. (1986) are of particles with diameters above (roughly) 170 nm, which doesn't indicate the correlation would hold for the total particle number concentration. Also Hegg and Russell (2000) have number concentration measurements only for particles with diameters larger 120 nm, and these measurements are made in marine environments.

We have added to the manuscript two references showing poor correlations between PM and PN, and reformulated the sentence in question as:

P2, lines 16-20: "Because the particles in different size ranges originate from different sources and atmospheric processes impact them differently, the particle number (PN) concentrations and particle mass concentrations (PM, e.g. $PM_{2.5}$ describing mass concentration of particles with $d_p < 2.5$ μm) are often poorly correlated even if considering only stationary measurements (e.g. Rodriguez et al., 2007; Rodrigues and Cuevas, 2007)."

*4) Page 3, lines 1-3 – Explain what you mean by neglecting the warming from BC due to their ability to form cloud droplets. This is an important point, but it is poorly described here.*

Neglect was an improper choice of word here. We have modified the sentence and added the citation later pointed out by the referee:

P3, lines 1-4: "…e.g. depending on the initial sizes and atmospheric growth of black carbon particles, the net-warming effect of the BC-rich particles can gradually change, either partly or entirely, and become net-cooling when the particles start to act as CCN and form cloud droplets (e.g. Chen et al., 2010)."

*5) Page 3, lines 3-5 – Discussions of the impact of reducing sulphur emissions started at least as far back as 1989 (Wigley, Nature).*

This reference is added (p3, line6).

*6) Page 3, lines 5-8 – Please re-write this sentence so the meaning is clear.*

The sentence rewritten as:

P3, lines 7-10: "However, the changes in aerosol-cloud interactions have been so far (if not ignored) assessed by assuming similar relative changes in particle mass and number emissions, which leads to incorrect results if the actual size distributions of emitted

particles change."

*7) Page 3, line 10 – New particle formation can result from primary emissions. Just refer to this as particle nucleation in the atmosphere.*

We have responded to comment 7 below, together with the response to comment 8.

*8) Page 3, Lines 12-15 – Another sentence needing clarification. I believe you intend this as an indication of particle formation during direct emissions, but it follows the sentence about nucleation in the atmosphere without connecting with direct emissions.*

We don't want to use primarily the term "nucleation", because it is not necessarily correct term for the new particle formation (if the formation occurs in kinetically limited, energetically barrierless process, the term nucleation is incorrect). On the other hand, we want to express that (regional scale) atmospheric and (small scale) combustion plume new particle formation are in principle the same process. We have modified the sentences pointed out by the referee and added the explanation also to Sect. 2.2. as follows:

P3 lines 12-22: "New particle formation (i.e. nucleation) produces particles with diameters below 2 nm (0.002 µm) from vapours such as sulphuric acid, organic vapours and nitrogen containing bases. This can happen both during regional scale atmospheric new particle formation events and at a smaller scale, for example in combustion plumes, when vapours suddenly cool immediately upon their introduction to ambient air. In this work, the latter, particles formed during the initial cooling and rapid dilution after the vapours are emitted to atmosphere, are also considered primary particles in addition to those emitted directly in particle phase. Somewhat larger UFP particles, still in nucleation mode size range, are formed e.g. in new particle formation processes occurring already before they are emitted to the atmosphere and thus producing cores for cooling vapours to condense on (e.g. Rönkkö et al., 2007; Lähde et al., 2010)."

P7 lines 5-10: "The emission factors and emissions described both in TNO work and in this study include both the particles emitted to atmosphere directly in particle phase, as well as those formed from vapours immediately after the emission during the rapid cooling and dilution of the exhausts. We consider here particles of both these types as primary particles. The uncertainties related to the emission factors in terms of particles formed immediately after the emissions are discussed in Sect. 4.1."

*9) Page 3, line 32 – Comment 3 above relates to this also.*

Responded above for comment 3).

*10) Both Amann references on pages 4 and 5 – The first refers to EMS and the second to TSAP. Both of these need explanations.*

The first, referring to the EMS article, is given as a reference for the GAINS model, which should be clear from how it is positioned in the text. For the second one, TSAP, we have added explanation:

P5, lines 9-13: "The GAINS model has been to support the Commission in the review of the Thematic Strategy on Air Pollution (TSAP; European Commission, 2005) and its related legal instruments on ambient air quality and national emission ceilings through modelling of emission baselines and scenarios for different policy options and their related impacts (Amann et al., 2013)."

*11) Page 6, line 26 – define TNO.*

Done.

*12) Page 6, line 28 to page 7, line 3 - You refer to sizes and sectors using "i", which is confusing. In equation 1, "i" is used to refer to a region, whereas in equation 2 it refers to a size class. Make it easier.*

Size classes are now referred with $n$.

*13) Page 6, Line 27 – Add a sentence or two to indicate what factors are used to distinguish primary emissions without influence from secondary processes.*

We have added the following sentence:

P7 lines 5-10: "The emission factors and emissions described both in TNO work and in this study include both the particles emitted to atmosphere directly in particle phase, as well as those formed from vapours immediately after the emission during the rapid cooling and dilution of the exhausts. We consider here particles of both these types as primary particles. The uncertainties related to the emission factors in terms of particles formed immediately after the emissions are discussed in Sect. 4.1."

*14) Page 7, lines 4-6 - EFpn:S? Is it intended as a plural of EFpn?*

Yes, $EF_{PN}$:s stands for plural of $EF_{PN}$.

*15) Page 8, line 23 – "New PSDs were" or "A new PSD was"?*

Corrected, p9 line 6; "A new PSD was…"

*16) Page 9, lines 3-5 - By black carbon mode, I assume you mean "pure" black carbon mode. Please clarify in the text.*

Sentence clarified:

P9 lines 17-20: "Two different size distributions were determined, one for the whole particles in black carbon mode ($BC_{mode}$), which considers both the black carbon cores and the condensed material on them, and one for the black carbon cores of these particles ($BC_{core}$). "

*17) Page 9, lines 9-14 - The Sorensen et al. work focussed on the morphology of larger particles. Their techniques and interest limited the discussion of smaller particles, and*

*indeed their micrographs indicate BC particles smaller than 50 nm. See Liggio et al. (Environ. Sci. Technol. 2012, 46, 4819−4828). It may be that your modal representation includes sufficient BC at sizes smaller than 50 nm, but the discussion needs to be a little more objective on this point.*

It is correct that our choice for minimum GMD for BC modes is a rough estimate and may exclude some BC-related modes from some sources, e.g. gasoline vehicles. We inspected the main sources with GMDs below but close to 50 nm and the results would not be much different if the limiting GMD was set e.g. to 40 nm. However, this does not mean that our approach is perfect, but we find it is something we can improve in the next version of emission factors. Accordingly, we added the reference cited by the referee and modified the text as:

P9 lines 24-32: "Of the combustion sources, only the modes with geometric mean diameters (GMD) equal to or above 50 nm were taken as black carbon modes. This rough estimate for a minimum GMD was chosen, because the agglomeration in BC formation produces a roughly lognormal mode and we assumed that would not form particles in the smallest size ranges of the modes with GMD below 50 nm (Sorensen et al., 1996; Kholghy et al., 2013). This assumption seems reasonable for diesel-fuelled vehicles, but might not be valid for gasoline-fuelled vehicles (Liggio et al., 2012). However, as the global emissions from diesel-fuelled vehicles are found to dominate the transport emissions, we will leave the further improvements on defining the black carbon modes to future studies."

We also added the following sentences to the end of Sect. 3.4.1.:

P17 line 30 - P18 line2: "It is to be noted that the method of defining the source-specific BC modes was approximate, as discussed in Sect. 2.3, and some of the sub-50 nm particles here defined as non-BC particles might in reality have a BC core. Even though this possible underestimation of smaller BC particles is unlikely to concern the diesel emissions (Liggio et al., 2012), which is the main source for BC number emissions, the black carbon size distributions from other sources should be assessed in more detail in future."

*18) Page 9, lines 18-21 – There has been significant debate as to what constitutes primary OC. This should be reflected in the response to comment 13 above.*

In these estimates we made the rough assumption that all OC in the studied size range is formed through condensation during the initial cooling of the exhaust gases. In reality, there is also a small fraction of solid $PM_{OC}$. For example, organic particles can also be emitted as solid particles if the combustion conditions are poor. This solid $PM_{OC}$ is always part of the primary OC independent of what the measurement method defines on cooling or dilution. It is however, a relatively small fraction of the total $PM_{OC}$ if some dilution and cooling are taken into account. For example, for conventional woodstoves, one of the most important categories in Europe, the average solid particle emission factor is 150 g $GJ^{-1}$ (range 49–650) – this includes a small fraction solid $PM_{OC}$ – whilst the average of the dilution tunnel measurements, that include both solid and condensable

particles, is 800 g $GJ^{-1}$ (range 290–1932). (Denier van der Gon et al., 2015, Nussbaumer et al., 2008)  The gap between the 150 and the 800 g $GJ^{-1}$ is entirely due to condensable, non-solid OC showing that this is by far the dominant $PM_{OC}$ fraction.

*19) Page 9, line 30 – "except for BC formed through condensation". This is confusing. I assume you mean PM1 minus the BC component. Please clarify.*

Modified as:

P10, line 16: "for the case of all $PM_1$, except for BC, assumed to be formed through condensation"

*20) Page 18, lines 23-24 – "are smaller"*

Modified: P19 lines 23-25: "In Europe and Northern America, the overall uncertainties, even though significant in absolute values, are smaller in comparison to the other continents, both in terms of current and future emissions."

*21) Page 19, line 19 – ". . . the primary emissions of . . ."*

Done (p20, line 21).

*22) Page 21, line 17 – "primary emissions"*

Done (p22, line 25).

*23) Page 21, line 21 – what do you mean by aerosol formation: secondary processes; nucleation?*

We mean both vapours condensing on the pre-existing particles and nucleation (referred to as new particle formation, see answer to comments 7 and 8). Clarified in the text:

P22, line 30 – P23, line 2: "…in the areas of low anthropogenic primary emissions the natural emissions and atmospheric aerosol formation (both in terms of vapours condensing on pre-existing particles and formation of new particles from vapours) play a relatively more important role…"

*24) Page 22, lines 1-15 – I am surprised that you have chosen to focus only on the warming aspect of BC, when your work seems suited to address the question about how much BC affects the number distributions and therefore the number concentrations of CCN; see Chen et al 2010, GRL. Estimating how the total number concentrations compare with the BC number concentrations from Figures 6 and 7 is very difficult, and in Figures 8 and 9 you only refer to BC mass. I would like to see a plot comparing the average number distribution from Figure 6 and the average BC mode and core number distributions from Figure 7. Of course there are still particles formed from secondary processes to be considered, but more knowledge of the importance of BC to primary emissions would be immensely useful.*

We did not intend to discuss only the warming aspect of BC, the changes in FP (roughly CCN) number emissions was discussed in the same sentence. However, the connection between BC mass and FP number emission was not pointed out. Thus, we have added the following:

P23, lines 18-20: "The predicted changes in BC mass emissions and PN emissions suggest that, even though the BC particles can act as CCN after atmospheric aging (Chen et al., 2010), the overall global decrease in BC mass emissions does not lead to similar decrease in number emission of FP."

It is true that the comparison between BC and non-BC particles was not easy enough from the original figures. Thus, we have substituted figure 7c with the figure the reviewer asked for, i.e. PSD of total global PN emissions and total global number emissions particles containing BC core. We found that the original Fig. 7c was so similar to 7b that the existing explanation of the difference was enough to describe it (P17, lines 24-29: "The difference between the assumptions of the composition of the coating of BC cores, i.e. the choice between coating including only OC and coating including all $PM_1$ except BC, was significant only in industrial combustion emissions, for which the BC core mode shifted to much smaller sizes (from ~100 nm to 30-40 nm) when assuming all $PM_1$ is condensed on BC cores.").

References:

Denier van der Gon, H. A. C., Bergström, R., Fountoukis, C., Johansson, C., Pandis, S. N., Simpson, D., and Visschedijk, A. J. H.: Particulate emissions from residential wood combustion in Europe – revised estimates and an evaluation, Atmos. Chem. Phys., 15, 6503-6519, doi:10.5194/acp-15-6503-2015, 2015.

Nussbaumer, T., Czasch, C., Klippel, N., Johansson, L., and Tullin, C.: Particulate Emissions from Biomass Combustion in IEA Countries, Survey on Measurements and Emission Factors, International Energy Agency (IEA) Bioenergy Task 32, Zurich, 2008